# Cellular In Vitro Responses Induced by Human Mesenchymal Stem/Stromal Cell-Derived Extracellular Vesicles Obtained from Suspension Culture

**DOI:** 10.3390/ijms25147605

**Published:** 2024-07-11

**Authors:** Ingrid L. M. Souza, Andreia A. Suzukawa, Raphaella Josino, Bruna H. Marcon, Anny W. Robert, Patrícia Shigunov, Alejandro Correa, Marco A. Stimamiglio

**Affiliations:** 1Laboratory of Basic Biology of Stem Cells (Labcet), Carlos Chagas Institute, Fiocruz, Curitiba 81350-010, PR, Brazilandreia.suzukawa@fiocruz.br (A.A.S.); bruna.marcon@fiocruz.br (B.H.M.); anny.robert@fiocruz.br (A.W.R.); patricia.shigunov@fiocruz.br (P.S.); 2Albert Einstein Israelite Hospital, São Paulo 05652-900, SP, Brazil; 3Confocal and Electronic Microscopy Facility (RPT07C), Carlos Chagas Institute, Fiocruz, Curitiba 81350-010, PR, Brazil

**Keywords:** extracellular vesicles, adipose tissue-derived mesenchymal stem/stromal cells, wound repair, angiogenesis, macrophage polarization

## Abstract

Mesenchymal stem/stromal cells (MSCs) and their extracellular vesicles (MSC-EVs) have been described to have important roles in tissue regeneration, including tissue repair, control of inflammation, enhancing angiogenesis, and regulating extracellular matrix remodeling. MSC-EVs have many advantages for use in regeneration therapies such as facility for dosage, histocompatibility, and low immunogenicity, thus possessing a lower possibility of rejection. In this work, we address the potential activity of MSC-EVs isolated from adipose-derived MSCs (ADMSC-EVs) cultured on cross-linked dextran microcarriers, applied to test the scalability and reproducibility of EV production. Isolated ADMSC-EVs were added into cultured human dermal fibroblasts (NHDF-1), keratinocytes (HaCat), endothelial cells (HUVEC), and THP-1 cell-derived macrophages to evaluate cellular responses (i.e., cell proliferation, cell migration, angiogenesis induction, and macrophage phenotype-switching). ADMSC viability and phenotype were assessed during cell culture and isolated ADMSC-EVs were monitored by nanotracking particle analysis, electron microscopy, and immunophenotyping. We observed an enhancement of HaCat proliferation; NHDF-1 and HaCat migration; endothelial tube formation on HUVEC; and the expression of inflammatory cytokines in THP-1-derived macrophages. The increased expression of TGF-β and IL-1β was observed in M1 macrophages treated with higher doses of ADMSC-EVs. Hence, EVs from microcarrier-cultivated ADMSCs are shown to modulate cell behavior, being able to induce skin tissue related cells to migrate and proliferate as well as stimulate angiogenesis and cause balance between pro- and anti-inflammatory responses in macrophages. Based on these findings, we suggest that the isolation of EVs from ADMSC suspension cultures makes it possible to induce in vitro cellular responses of interest and obtain sufficient particle numbers for the development of in vivo concept tests for tissue regeneration studies.

## 1. Introduction

Mesenchymal stem/stromal cells (MSCs) are a class of multipotent somatic adult stromal cells that reside in a variety of tissues such as the adipose tissue, dental pulp, umbilical cord, and bone marrow [1,2]. The regenerative, protective, and anti-inflammatory properties of MSCs are well documented [3,4] and make them appropriate candidates for regenerative medicine and tissue engineering applications [1,5]. However, the use of MSCs in clinical treatment still has major negative issues that require solving, such as donor-dependent variability, cellular viability, and aberrant differentiation [4,6]. MSCs have been described to have important roles in tissue regeneration and wound closure and repair, reducing inflammation [7,8], enhancing angiogenesis [9], and regulating ECM remodeling [8]. In studies in which these cells have been injected locally into or around the wound area, the treatment effects have been limited by cell retention at the site of wound [10]. MSCs secrete a variety of compounds such as cytokines, growth factors, and extracellular vesicles that influence nearby surrounding cells and alter the tissue microenvironment and have a more efficient paracrine role in inducing tissue repair than direct MSCs therapy [11,12]. The paracrine activity of MSCs plays a crucial role in their regenerative capabilities [13,14,15].

MSC-derived extracellular vesicles (MSC-EVs) are known to induce tissue regeneration. Fractions of EVs include exosomes, microvesicles, and apoptotic bodies [16]. These EVs carry a diverse cargo comprising nucleic acids (DNA, RNA, mRNA, and miRNA), enzymes, and pro- and anti-inflammatory cytokines [17]. Released into the extracellular space, the EVs facilitate cell-to-cell communication, exerting significant influence on the physiological and pathological conditions of recipient cells [18,19]. Prior research has identified over 150 miRNAs and 850 unique proteins in MSC exosomes (MSC-exos) cargo, impacting target cells. MSC-exos carry a variety of bioactive molecules such as cytokines and growth factors, and among them are the transforming growth factor beta-1 (TGF-β1), IL-6, IL-10, and the hepatocyte growth factor (HGF), which are involved in immune modulation and the homing of immune cells [13,20,21,22]. Additionally, they contain VEGF, EMMPRIN, and MMP-9, which contribute to angiogenesis and tissue repair [13]. Moreover, MSC-exos express MSC markers (CD29, CD44, CD90, and CD73), enhancing their affinity for residing in injured and inflamed tissues [1,21]. In addition, MSC-exos have some advantages, including the ability to cross blood–brain barriers and capillaries [1,17], non-oncogenicity, high stability, and cell and tissue-specific homing; also, they do not induce vascular obstruction [23,24] and have low immunogenicity [1].

In skin wounds, regeneration is a challenging process because the cells are in an inflamed environment in which the secretion of pro-inflammatory cytokines such as TNF-α, IL-6, IL-1β, and IL-17 occurs, as well as the secretion of inducible nitric oxide synthase (iNOS) and reactive oxygen species (ROS); tissue degradation; and a decrease in cell proliferation capacity [25,26]. For regeneration to occur properly, it is necessary to control inflammation, to induce macrophage phenotype polarization from M1 to M2, and to induce skin cell proliferation, migration, and the secretion of extracellular matrix components [26].

During response to stimuli, macrophages can be differentiated into two distinct functional phenotypes: classically activated pro-inflammatory macrophages (M1), which express cytotoxic and pro-inflammatory cytokines such as IL-1, TNF-α, IL-1β, and IL-6, promoting microbicidal effects [26], and alternatively activated macrophages (M2), which express anti-inflammatory cytokines such as IL-4, IL-13 [27,28], IL-10, TGF-β, placental growth factor, and other cytokines and growth factors that induce reduction in inflammation and promote the cell proliferation and remodeling of the extracellular matrix [26,29]. The wound microenvironment strongly influences the polarization and functional heterogeneity of the macrophages [30]. The imbalance of M1 and M2 induces an increase in pro-inflammatory cytokines that inhibit tissue repair, as in chronic diabetic wounds, in which macrophages secrete an exorbitant amount of IL-1β, TNF-α, MCP-1, MMP-9, FGF-2, IL-17, ROS, and iNOS [25,31,32]. Hence, during regeneration, the environment’s characteristic is altered to a pro-regenerative state in which extracellular matrix secretion and deposition, fibroblast proliferation, the secretion of VEGF by endothelial cells resulting in angiogenesis, and the secretion of anti-inflammatory cytokines occurs [33]. Additionally, stem cells and immune cells exert paracrine effects on each other through EVs [34]. Stem cell EVs preferentially accumulate in injury sites to inhibit the pro-inflammatory response of immune cells [34,35] and to inhibit their proliferation and activation [34,36,37]. On the other hand, EVs derived from immune cells also exert their effects on stem cells, promoting their recruitment and migration to the injury site [34,38]. Hence, the bidirectional signaling of MSC-EVs and immune cells EVs is necessary for proper wound healing.

The immunomodulatory and pro-regenerative potential of MSC-EVs makes them an important alternative for application in regenerative medicine. However, their production has some challenges in terms of increasing their quantity and yield. In order to overcome the challenges of small-scale EV isolation from 2D cultures, microcarrier suspension cultures have been developed to isolate large-scale EVs for future use in the clinic. In this prospection, many studies developed microcarriers for the growth of anchorage-dependent mammalian cells with different features that can affect cell attachment, spreading, growth, and differentiation. The surface of microcarriers is diverse and can contain functional groups such as positively charged tertiary, quaternary or primary amines, gelatin, collagen, and other extracellular matrix (ECM) proteins and peptides (e.g., RGD) [39]. It is crucial that the microcarriers have positive charges to fix the cells through electrostatic forces, as they are negatively charged. Furthermore, their surfaces can contain positively charged materials such as cross-linked dextran or spheres derived from polyacrylamide with tertiary amines [40,41,42]. More recently, microcarriers have been used for large-scale cell expansion in order to harvest cells, supernatants, and EVs for pharmaceutical products and therapies development.

In this study, we explore the putative regenerative potential of ADMSC-EVs derived from the suspension cultures of ADMSCs grown on cross-linked dextran microcarriers (Cytodex-1). We investigated their effects on various skin cell types including fibroblasts and keratinocytes, as well as macrophages and endothelial cells. Our analysis addressed proliferation, “wound closure”, angiogenic stimulation, and the modulation of pro- and anti-inflammatory cytokine expression in THP-1-derived macrophages treated with ADMSC-EVs. Our findings revealed enhanced keratinocyte proliferation, increased migration in both keratinocytes and fibroblasts, and augmented angiogenic properties in HUVEC cells following ADMSC-EV treatment. Furthermore, increased TGF-β and IL-1β expression was detected in M1 macrophages treated with higher concentrations of ADMSC-EVs. Also, the expression of pro-inflammatory cytokine TNF-α and anti-inflammatory IL-10 were non-significant in all groups of macrophages treated with ADMSC-EVs independently of EVs concentration. Hence, EVs from microcarrier-cultivated ADMSCs appear to be suitable to induce different cell responses and should be tested for regeneration processes.

## 2. Results

### 2.1. ADMSCs Cultured on Cytodex-1 Are Viable and Do Not Present Alterations in Morphology and Immunophenotype

In order to obtain a large quantity of cells and, consequently, secreted EVs, cell cultivation was set on cytodex-1 under agitation in Erlenmeyer flasks. The ADMSCs seeded on microcarriers remained adhered to the cytodex-1 dextran matrix throughout the cultivation period under agitation, maintaining their spreading fibroblast-like morphology characteristics (Figure 1A). In parallel, ADMSCs cultured over 4 days under these conditions were immunophenotyped and presented typical mesenchymal markers. The population of ADMSCs cultured on cytodex-1 showed positive staining in more than 95% of the following mesenchymal cells markers: CD90, CD105, CD73, and CD140b (Figure 1B). Cell populations were essentially negative for CD45, CD34, CD11b, HLA-DR, and CD19 antigens (Figure 1C), remaining comparable to those cultured in standard 2D plates.

ADMSCs cultured on cytodex-1 were further analyzed regarding their morphology. The staining of beta-actin (Figure 2A), F-actin (Figure 2B), beta-tubulin (Figure 2C), and vimentin (Figure 2D) showed that an ADMSC’s cytoskeleton remained with the typical spindle shaped morphology. The secretion of the extracellular matrix protein fibronectin by ADMSCs was also observed (Figure 2E) in the cytodex-1 surface. LIVE/DEAD assays showed that cells adhered to cytodex-1, even when maintained without fetal bovine serum, remaining alive and viable (in green, Figure 2F). The results observed so far show that the cultivation of ADMSCs on cytodex-1 microcarriers, in suspension and under agitation does not cause significant cell death and allows the establishment and expansion of cell culture on a large-scale, at least under the tested conditions.

### 2.2. Extracellular Vesicles Isolated from ADMSCs Cultured in Cytodex-1 Retain Their Characteristics Regarding Size and Phenotype

EVs isolated from ADMSCs cultivated on cytodex-1 were analyzed in terms of their morphology and molecular profiles. Considering the isolation of three independent EV samples, it was observed that ADMSCs cultivated on cytodex-1 (in a condition of 6 × 10^6^ cells to each 1 g/L cytodex-1) secrete on average (±standard deviation) 2 × 10^10^ ± 0.28 × 10^10^ VEs/mL, which represents 430 ± 23.2 μg/mL of protein, and have an approximate diameter of 130 ± 21.2 nm (Figure 3A). ADMSC-EVs evaluated by transmission electron microscopy (TEM) showed morphological characteristics compatible with exosomes (Figure 3B) and presented specific general markers such as CD63, CD9, and TSG101 (Figure 3C). EVs isolated from ADMSCs culture on cytodex-1 has shown to be consistent, with uniform size, quantity, and protein concentration in all isolations.

### 2.3. EVs from ADMSCs Cultured on Cytodex-1 Are Properly Taken up by Cells and Induce Keratinocyte Proliferation

After EVs characterization, the internalization of these vesicles was evaluated in skin and endothelial cells. Keratinocytes (HaCat), fibroblasts (NHDF-1), and endothelial cells (HUVEC) were capable of internalizing red-stained ADMSC-EVs (EVs-PKH26). In Figure 4, the red dots of EVs-PKH26 in the cells can be seen. Comparatively, NHDF-1 cells (Figure 4B) and HUVECs (Figure 4C) were able to internalize greater amounts of ADMSC-EVs than HaCat keratinocytes.

To evaluate the proliferation of cells treated or not treated with ADMSC-EVs, Ki67 nuclei staining was performed. A significant increase in the proliferation of HaCat keratinocytes was observed when treated with ADMSC-EVs for 48 h in comparison to untreated cells (Figure 5A,D). Fibroblasts (NHDF-1) and endothelial cells (HUVEC) did not increase proliferation after treatment with ADMSC-EVs for 48 h in comparison to untreated cells (Figure 5B–D).

### 2.4. ADMSC-EVs Induced Migration of Fibroblasts and Keratinocytes

Aiming to evaluate the migration induction potential of ADMSC-EVs in fibroblasts and keratinocytes, a scratch wound assay was performed using NHDF-1 and HaCat cells. Cells were photographed and evaluated at 0, 6, 12, 18, and 24 h with or without starvation (maintained in a medium depleted of FBS) (Appendix A). The scratch area was reduced over time in all groups, being almost absent at 24 h. Cell migration and closure area after 24 h were observed in groups treated with ADMSC-EVs in both cell types, NHDF-1 (Figure 6A) and HaCat (Figure 6B). In control groups maintained in medium supplemented with FBS, the migration effect of ADMSC-EVs could not be detected because migration was most probably masked by FBS (Appendix A).

### 2.5. ADMSC-EVs Induced Angiogenesis

To evaluate angiogenesis induction by ADMSC-EVs an in vitro tube formation assay was performed using HUVECs. Endothelial cells treated with ADMSC-EVs developed a more branched network of tube-like structures (Figure 7A, upper panel) when compared to non-treated HUVECs (Figure 7A, lower panel). Upon treatment with ADMSC-EVs, endothelial cells appeared to be well distributed along tube-like structures, without clustering at nodes as in the untreated group. It was also observed that HUVECs treated with ADMSC-EVs presented significant differences in number of branches and nodes, as well as in the length of the branches, when compared to non-treated cells (Figure 7B). Other features such as number of extremities, junctions, and segments were not statistic different between groups. HUVEC treated with ADMSC-EVs showed expression of endothelial marker CD31 (Figure 7C, upper panel) and F-actin marker (Figure 7C, lower panel). In HUVEC maintained without FBS, the organization of tubular structures was impaired; however, in cells treated with ADMSC-EVs, it was possible to observe the beginning of tubular organization that had not yet been completely formed (Appendix A).

### 2.6. ADMSC-EVs Induced Cytokine Expression in THP-1-Derived Macrophages

To investigate the potential of ADMSC-EVs to modulate immune activity, THP-1 cells were induced to differentiate into macrophages. Unstimulated M0 differentiated macrophages and both M1-like and M2-like macrophages were treated or not treated for 72 h with ADMSC-EVs. The differential gene expression of inflammatory (IL-1β and TNF-α) and anti-inflammatory (IL-10 and TGF-β) cytokines was evaluated by RT-qPCR. To see phenotypic and morphological changes during treatment with PMA, ADMSC-EVs and cytokines, cells were photographed at days 2, 4, and 6 (Appendix A). It was seen that, on day 2 of treatment, all cells were adhered but less spread out than on day 6 (Appendix A). The treatment of M0, M1-like, and M2-like macrophages with the lower dose of ADMSC-EVs (10 μg/mL) induced a non-significant expression of pro-inflammatory and anti-inflammatory cytokines in general (Appendix A). Independently of the ADMSC-EVs concentration, EV-treatment did not induce significant levels of anti-inflammatory IL-10 in any of macrophage groups (Figure 8A and Appendix A). EV-treated M0 and M2-like macrophages exhibited the non-significant expression of pro-inflammatory cytokines IL-1β and TNF-α independently of ADMSC-EV concentration, when compared to non-treated cells (Figure 8A and Appendix A). Only M1-like macrophages incubated with 30 μg/mL of ADMSC-EVs showed significant expression levels of IL-1β and TGF-β cytokines (Figure 8A). The analysis of the relative cytokine expression of all macrophage groups treated with 10 μg/mL of ADMSC-EVs showed a non-significant ratio between anti- and pro-inflammatory cytokines (Appendix A). Besides that, the analysis of relative cytokine expression showed that M1-like macrophages incubated with 30 μg/mL of ADMSC-EVs exhibited a higher ratio of pro-inflammatory TNF-α expression to anti-inflammatory IL-10 and a higher ratio of pro-inflammatory IL-1β to anti-inflammatory IL-10 (Figure 8B).

## 3. Discussion

The use of MSCs and their biological byproducts in regenerative medicine has expanded in recent decades. Due to high demand, it is necessary to use a large-scale culture of MSCs. Hence, many strategies for MSCs large-scale culture have been developed, among them, the use of microcarriers in suspension cultures [39,43,44,45,46,47,48,49,50,51,52,53,54,55,56]. Therapies with ADMSCs utilize their regenerative potential in tissue repair. Studies using animal models and in vitro cell cultures reported that MSCs can improve tissue regeneration efficiency, reduce the level of inflammation, and promote faster wound closure [57,58,59,60,61,62,63]. Furthermore, ADMSC-EVs or ADMSC exosomes have been described with different and important roles in regenerative cell-free therapies. Many studies showed that ADSCs-EVs can reduce inflammatory response and has regenerative effects [57,58,59,60,61,62,63]. In this work, we investigated the potential of ADMSC-EVs isolated from the suspension cultures of ADMSCs to modulate cell behavior.

Firstly, we observed that ADMSCs cultured on cytodex-1 under agitation and in serum-free medium remained adhered to microcarriers with preserved morphology and phenotypic markers similar to ADMSCs cultured on scaffold [64] or in 2D culture flasks. In addition, we observed that ADMSCs cultured on cytodex-1 actively secreted extracellular matrix proteins (i.e., fibronectin) on the surface of the microcarrier. EVs isolated from these cell cultures did not show morphological or size alterations, as well as presenting characteristic EV surface markers, similar to EVs isolated from other sources of mesenchymal cells [65]. Hence, cytodex-1 is a suitable microcarrier for ADMSCs large-scale growth. These results are in agreement with other studies showing that ADMSCs cultured on different microcarriers, in xeno- and serum-free medium under dynamic conditions, do not degenerate and maintain similar characteristics to those grown in traditional static 2D systems [66]. Previous studies have used dextran-coated cytodex-1 for general mesenchymal stem cells [42,46,47,48,49] and different specific types of mesenchymal stem cells expansion and/or differentiation such as bone marrow mesenchymal stem cells (BMMSCs) [50,51,52,53,54,55], ADMSCs [56], Wharton’s jelly mesenchymal stem cells (WJ-MSC) [57], dental pulp stem cells (DPSCs) [58], and others. Furthermore, it was reported that ADMSCs can preserve their undifferentiated status and be less inflammation-prone in dynamic 3D conditions [66]. It is also known that MSC-EVs isolated from 3D conditions are more effective than EVs obtained from monolayer cell culture in inducing angiogenesis and reducing apoptosis and inflammation through the increase in anti-inflammatory cytokines and reduction in pro-inflammatory cytokine expression [67]. Hence, the culture of ADMSCs on microcarriers might be a path not only for large-scale cell growth, but for the large-scale production of their biological byproducts.

Despite that, we did not see the significant proliferation of fibroblasts and HUVEC, and we observed increased proliferation on HaCat cells treated with ADMSC-EVs when compared to non-treated group. In this case, our findings are in concordance with many other works showing that MSC-EVs induce HaCat proliferation. Ren and colleagues (2019) reported that ADMSC-EVs isolated from the 2D culture promoted the in vitro proliferation and migration of fibroblasts, HaCat keratinocytes, and HUVEC [68]. Other findings showed that ADMSC-EVs promoted in vitro HaCaT keratinocyte proliferation and in vivo wound healing [69]. ADMSC-EVs are also reported to be rich in proangiogenic factors and inductors of the proliferation of endothelial cells [70]. Li and colleagues (2021) reported that ADMSC exosomes facilitated wound healing and reduced hypertrofic scars (fibrosis) in mice [71]. Interestingly, in a mouse burn wound model, ADMSC-EVs accelerate wound closure and increased epithelial thickness, collagen deposition, and neovascularization [72].

In addition, through an in vitro scratch assay, we detected a significant migration stimulus on EVs-treated keratinocytes and fibroblasts. It is important to note that our assay design involved cell starvation with the removal of FBS from the culture, because its presence masked the effect of ADMSC-EVs on this process. It is well established that MSC-EVs enhance restoration processes such as re-epithelialization, neovasculogenesis, proliferation, and cell migration to the injured site and triggers the production of collagen 1 and 3, fibronectin, and extracellular matrix components [16,73,74,75,76,77]. Previous work showed that ADMSC exosomes promote ECM remodeling in cutaneous wound repair by regulating the ratios of collagen type III: type I, TGF-β3:TGF-β1, and MMP3:TIMP1 via the ERK/MAPK signaling pathways [78]. Other findings showed that MSC exosomes induce the expression of extracellular matrix proteins and increase cell migration [79]. These findings can explain migration mechanisms that might be occurring in the NHDF-1 and HaCat cells tested in our work.

Wang and coworkers (2023) showed that ADMSC exosomes induce the in vitro proliferation and migration of wound cells and increase the number of blood vessels (vascularization) [80]. Furthermore, ADMSC exosomes can reduce ROS damage, induce angiogenesis, and enhance proliferation and collagen accumulation in diabetic wounds [81]. Interestingly, we observed that endothelial cells treated with ADMSC-EVs presented significant differences in the number of branches and nodes and in branch length when compared to non-treated cells in endothelial tube formation assay. In addition, HUVEC treated with ADMSC-EVs developed well-architected tubes and showed the expression of the CD31 endothelial marker. Steps of angiogenesis such as tube and vessel formation are crucial processes for the irrigation of the regenerated tissue. Many studies have demonstrated that ADMSC-EVs induce angiogenesis in wound repair. It is known that treatment with ADMSC exosomes in diabetic mouse model induce the elevation of angiopoietin 1 (ANG1), fetal liver kinase-1 (FILK1), and VEGF and reduction in angiogenesis inhibitors angiogenin-1 (VASH1) and thrombopoietin 1 (TSP1), enhancing angiogenesis [82]. Other findings showed that ADSCs-exosomes with high Nrf2 expression induced endothelial progenitor cell proliferation and angiogenesis by improving the phosphorylation levels of SMP30, VEGF, and VEGFR2 [59,83]. In addition, EVs from human umbilical cord mesenchymal cells (UCMSC-EVs) are known to accelerate wound healing and angiogenesis and increase the expression of VEGF and TGF-β1 [84].

Inflammatory response in tissue regeneration must be balanced. For example, the skin healing process starts just after tissue injury. Initially, the process of inflammation is necessary for pathogens and debris clearance, with the rapid production of pro-inflammatory cytokines (e.g., IL-1, IL-2, IL-6, IL-8, TNF-α, IFNs, and prostaglandins) and growth factors (TGF-β, EGF, PDGF, and FGF) at the site of injury [85]. These factors promote the migration of inflammatory cells into the wound environment. Then, the inflammation must be attenuated to avoid damage of the nearby healthy tissue. Hence, a polarization from pro-inflammatory M1 to anti-inflammatory M2 macrophages must occur in order to enhance wound healing. M2 macrophages secrete anti-inflammatory cytokines and growth factors and play important roles in wound healing [22]. MSC-EVs are known to play immunomodulatory roles that trigger effective wound healing [86]. Many studies reported that MSC-EVs induce the increase in anti-inflammatory cytokines such as IL-10 and TGF-β and the reduction in pro-inflammatory cytokines such as TNF-α and IL-1β, thus inducing M2 macrophage polarization [21,87,88,89,90]. However, in our work, we did not observe the significant increased expression of IL-10 and other anti-inflammatory cytokines. We observed the increased expression of pro-inflammatory cytokine IL-1β and the anti-inflammatory cytokine TGF-β in THP-1-derived M1-like macrophages treated with higher concentrations of ADMSC-EVs. Previous studies demonstrated that IL-1β has a positive role in wound healing: by inducing fibroblast and keratinocyte proliferation in a paracrine manner [91,92,93,94,95]; through the production and degradation of extracellular matrix proteins (e.g., collagens, elastins, fibronectins, and laminins); inducing fibroblast chemotaxis; and inducing keratinocytes to secrete VEGF, a key factor in neovascularization [95,96]. Unfortunately, we did not test the effect of cytokines secreted by macrophages on the proliferation of fibroblasts and keratinocytes, and further studies in this area are recommended.

It is known that the anti-inflammatory phenotype of MSCs must be activated by exposure to a specific environment (e.g., pro-inflammatory cytokines treatment) and that unprimed MSCs have their immunosuppressive effect reduced [97,98,99]. So, the immunosuppressive effect of MSCs requires the “licensing” of inflammatory factors. In an inflammatory environment (with higher concentrations of TNF-α and IFN-γ), MSCs are activated and inhibit T cell proliferation through the secretion of soluble factors such as IDO, PGE2, NO, TGF-β, HGF, and heme oxygenase (HO). Hence, activated MSCs turn to an immunosuppressive phenotype (MSC2) [99,100]. In the absence of a pro-inflammatory environment (with lower concentrations of TNF-α and IFN-γ), MSCs may exhibit a pro-inflammatory phenotype (MSC1) and enhance T cell responses by secreting chemokines (e.g., MIP-1α and MIP-1β, RANTES, CXCL9, and CXCL10) to recruit lymphocytes to sites of inflammation [99,101,102]. When displaying the MSC1 phenotype, the levels of immunosuppressive mediators such as IDO and NO are low [99]. In our model, MSCs were not primed in a pro-inflammatory microenvironment or stimulated with molecules that could trigger signaling for the secretion of anti-inflammatory EVs. Hence, our ADMSC-EVs had the weakest effects on M2 macrophage polarization and, consequently, on anti-inflammatory cytokine expression. In research by Zidan and colleagues (2021), EVs isolated from human MSCs derived from urine samples induced the increased expression of the activation marker CD69 on B cells and increased their proliferation in culture. These MSCs were cultured without inflammatory stimulus and the EVs obtained contained the presence of BAFF, APRIL, IL-6, and CD40L. On the other hand, in this same study, a significant suppression of the proliferation of activated T cells was demonstrated after treatment with EVs [103]. Considering this analysis, we suggest new studies with a pre-treatment of ADMSCs to induce the secretion of EVs with a stronger anti-inflammatory profile, as well as other assays to evaluate the immunomodulatory potential of the ADMSC-EVs. Regarding this, previous studies performed MSCs priming with pro-inflammatory cytokines to induce the secretion of EVs with anti-inflammatory and regenerative potential [104]. Pre-treatment of BMMSCs with melatonin induce the secretion of EVs that promote in vitro M2 macrophage polarization in murine RAW264.7, inhibit the pro-inflammatory response, and promote diabetic wound healing [105]. Furthermore, LPS-primed MSC-EVs transports microRNA let-7b, reducing TLR-4 expression and NF-κB activation, reducing pro-inflammatory signaling, and triggering signaling for wound healing and M2 macrophage polarization via the activation of the TLR4/NF-κB/STAT3/AKT pathway [106]. The MSC priming with pro-inflammatory stimuli or 3D culture also induced profound transcriptomic changes and enhanced the immunosuppressive potency of MSCs [107]. Microenvironmental stimuli in cell culture, including hypoxia [28] and the insertion of external cargo in VEs, such as BMP-2/VEGF-A loaded by electroporation [108] may induce EVs with anti-inflammatory profile and the potency to injury regeneration.

Regarding the TGF-β1 expression, our results are in accordance with the literature showing that ADSCs-exosomes could stimulate monocytes or macrophages to secrete this cytokine, stimulating fibroblast proliferation and promoting the production of type I collagen in diabetic wounds [59,109]. MSC-EV-treated macrophage RAW264.7 increased anti-inflammatory TGF-β1 cytokine expression, thus elevating miR-132 expression, which is involved in the induction of macrophage M2 polarization [110].

It is important to reinforce some experimental limitations of this work: (1) the lack of more than one biological replicate of ADMSCs to harvest EVs; (2) the lack of diversity in EV target cells, only one model of fibroblast, keratinocyte, and endothelial cells was used; (3) the THP-1 cell model limitations for reproducing a true macrophage response; and (4) the lack of analysis on EVs molecular content. Further investigations with suspension cultured ADMSC-EVs using primed ADMSCs are needed to overcome experimental gaps and unleash the more efficient pro-regenerative and anti-inflammatory potential of these EVs.

## 4. Materials and Methods

### 4.1. Standard Cell Culture Conditions and Reagents

Human dermal fibroblasts (NHDF-neo_21827, cat. 2509, lot 231340, C478818, Lonza, Basel, Switzerland), keratinocytes (HaCat from the Carlos Chagas Institute cell bank), endothelial cells from umbilical vein (HUVEC, cat. C2519AS, Lonza), and adipose-derived stem/stromal cells (ADMSCs) were cultivated in low passages. ADMSCs, NHDF-1, and HUVEC were used until maximum 10 passages and HaCat was used until 26 passages for experimental purposes. ADMSCs were obtained from liposuction of 2 different healthy donors and their use was in accordance with the guidelines for research involving human subjects and had the approval of the Ethics Committee of Fundação Oswaldo Cruz, Brazil (CAAE: 48374715.8.0000.5248). ADMSCs, fibroblasts NHDF-1, and HaCat were maintained in DMEM high-glucose (cat. 12100-046, Gibco) supplemented with 10% fetal bovine serum (FBS, cat. 12657-029, Gibco, Waltham, MA, USA), 1% l-Glutamine (cat. 21051-024, Sigma, St. Louis, MO, USA), and 1% Penicillin/Streptomycin (P/S, cat. 15140-122, Gibco). HUVECs were maintained in EBM-2 medium (cat. CC-3156, Lonza) supplemented with EGM^TM^ 2 MV SingleQuots^TM^ (cat. CC-4147, Lonza) containing 10% FBS (cat. CC-4102B, Lonza), hydrocortisone (cat. CC-4112B, Lonza), hFGF-B (cat. CC-4113B, Lonza), VEGF (cat. CC-4114B, Lonza), hEGF (cat. CC-4317B, Lonza), R3-IGF-1 (cat. CC-4115B, Lonza), ascorbic acid (cat. CC-4116B, Lonza), and Gentamicin-Amphotericin (GA-1000, cat. CC-4381B, Lonza). All cells were maintained in incubator with 5% CO_2_ at 37 °C. During supernatant harvesting for EVs isolation purposes, ADMSCs were maintained in DMEM medium without FBS to avoid the isolation of bovine vesicles together with ADMSC-EVs.

### 4.2. ADMSCs Culture on Microcarriers and Immunophenotyping

ADMSCs at a concentration of 6 × 10^6^ cells were mixed to 1 g/L of cytodex-1 (cat. 17548702, Cytiva, Amersham, UK) microcarrier beads and cultured in Erlenmeyer flasks with 20 mL of DMEM high-glucose supplemented with 10% FBS. Cells on microcarriers were maintained under agitation of 115 rpm in an atmosphere with 5% CO_2_ at 37 °C. Cells were maintained from passages 6–10 in all the culture procedures. Their growth was observed over 4 days through microscopy visualization. Cell immunophenotyping was assessed to confirm if cytodex-1 microcarriers would maintain ADMSCs phenotype throughout the culture. Cells were detached from microcarriers and passed through 40 μm pores cell strainers (BD, Falcon, Schaffhausen, Switzerland). Flow cytometry was applied for cell immunophenotyping using the classic MSC positive (CD90, CD105, CD73) and negative (CD11b, CD45, CD34) membrane markers as previously described [111].

The following antibodies were used: mouse IgG FITC (1:50, cat. 555748, BD); mouse IgG PE (1:50, cat. 555749, BD); mouse IgG APC (1:50, cat. 555751, BD); mouse anti-human CD90 FITC (1:5, cat. 11-0909-73, eBioscience, San Diego, CA, USA); mouse anti-human CD105 PE (1:50, cat. 12-1057-42, eBioscience); mouse anti-human CD73 APC (1:50, cat. 17-0739-42, eBioscience); mouse anti-human CD34 FITC (1:50, cat. 11-0349-42, eBioscience); mouse anti-human CD11b PE (1:50, cat. 12-0112-82, eBioscience); mouse anti-human CD45 APC (1:50, cat. 17-0459-73, eBioscience); mouse anti-human CD19 FITC (1:50, cat. 555412, BD); mouse anti-human HLA-DR APC (1:50, cat. MHLDR05, Life Technologies, Carlsbad, CA, USA); and mouse anti-human CD140b PE (1:5, cat. 558821, BD). The samples were analyzed in a FACSCanto II cytometer device (BD, Biosciences, Franklin Lakes, NJ, USA) at the Carlos Chagas Institute FIOCRUZ-PR flow cytometry facility, and results were analyzed using FlowJo version 10.8.1 software.

### 4.3. Cell Viability Evaluation

To analyze whether the absence of FBS, required during the supernatant harvesting for EVs isolation, or the physical features of suspension culture, such as agitation, cause cell damage and death, a LIVE/DEAD viability assay was performed. ADMSCs were cultured on cytodex-1 microcarriers for 4 days with DMEM high-glucose medium supplemented with FBS and then for more 3 days with same medium without FBS. Next cells were stained with LIVE/DEAD™ Viability/Cytotoxicity Kit for mammalian cells (cat. L3224, Invitrogen, Waltham, MA, USA) for 40 min at room temperature according to the manufacturer’s instructions. The samples were analyzed under a fluorescence Leica TCS SP5 confocal microscope at the Carlos Chagas Institute FIOCRUZ-PR microscopy facility.

### 4.4. ADMSCs Cytoskeleton and Extracellular Matrix Characterization

In order to evaluate the cell morphology and secretion of extracellular matrix on ADMSCs cultured in microcarriers, immunofluorescence assays were performed. For the characterization of cytoskeletal proteins, cells grown on microcarriers were fixed, permeabilized, and labeled with anti-β-actin 8H10D10 (1:100, cat. 3700, Cell Signaling, Danvers, MA, USA) or phalloidin A488 (1:400, cat. A12379, Invitrogen), anti-β-tubulin FITC (1:100, cat. 16230, Sigma), or anti-vimentin (1:100, cat. 550513, BD Biosciences). For the evaluation of extracellular matrix secretion, cells were labeled with anti-fibronectin (1:100, cat. 610078, BD Biosciences), and necessary secondary antibodies Alexa fluor 488 donkey anti-rabbit IgG (Invitrogen, cat. A21206), Alexa Fluor 488 Donkey anti-mouse IgG (Invitrogen, cat. A21202), and goat anti-rabbit IgG Alexa Fluor™ 546 (Invitrogen, cat. A-11010) were used. Images were achieved using a fluorescence Leica DMI6000B microscope and LAS AF software 3.0.

### 4.5. Extracellular Vesicles Isolation

After 4 days of ADMSC culture on cytodex-1 microcarriers (with approximately 80% of confluency), cells were washed twice with calcium- and magnesium-free buffered salt solution (BSS-CMF) and cultured on DMEM high-glucose medium without the supplementation of FBS. After 24 h of incubation, cells on microcarriers were centrifuged at 3100× *g* for 5 min. Then, the supernatant without microcarriers was harvested and transferred to a new tube. Microcarriers with cells were resuspended in fresh medium and redistributed in Erlenmeyer flasks. Supernatants were centrifuged at 700× *g* for 5 min at 4 °C to remove cell debris. Then, supernatants were harvested and centrifuged again at 4000× *g* for 20 min at 4 °C to remove apoptotic bodies. This process of harvest and centrifugation was repeated for 3 consecutive days. On the third day, approximately 216 mL of supernatants were combined and submitted to ultracentrifugation at 100,000× *g* for 1 h and 20 min at 4 °C in an ultracentrifuge Himac CP 80wx with P28S rotor (Hitachi, Tokyo, Japan) to pellet the EVs. After ultracentrifugation, supernatants were discarded and EV pellets were resuspended thrice with 200 µL of phosphate-buffered saline (PBS) under agitation to recover the highest amount of EVs. Then, all the resuspended EVs were ultracentrifuged again at 100,000× *g* for 2 h at 4 °C to eliminate possible contaminants from culture medium. By the end of ultracentrifugation, the supernatant was discarded and the EV pellets were resuspended in PBS. EVs were stored in an ultrafreezer at −80 °C until experimental procedures.

### 4.6. Nanotracking Analysis of EVs

All ADMSC-EVs samples were diluted 1:10 in PBS to a final volume of 1 mL and then analyzed and quantified using a Nanosight LM14C instrument (Malvern Instruments, Malvern, UK). The ideal measurement concentration was 20–100 particles/frame. The following settings were used: the camera level was increased until all particles were distinctly visible, not exceeding a particle signal saturation of 20% (cell line-derived EVs: camera levels between 11–13). The ideal detection threshold was determined to include as many particles as possible within a range of 10–100 red crosses. Also, blue crosses were limited to 5. Autofocus was adjusted so that indistinct particles were avoided. For each measurement, three 60 s videos were captured under the following conditions: 25 °C cell temperature; infusion rate at 1000 units; and syringe pump speed at 100 µL/s. After capture, the videos were analyzed by the NanoSight Analytical Software—NTA 3.4, Build 3.4.4, with a detection threshold of 2 or 3, in order to exclude unspecific non-EVs particles.

### 4.7. Scanning Electron Microscopy Imaging of ADMSCs

Cells on microcarrier samples were fixed with a solution of glutaraldehyde (2.5%) in sodium cacodylate buffer (0.1 M). Then, samples were washed with sodium cacodylate buffer (0.1 M), post-fixed (1% osmium tetroxide in 0.1 M sodium cacodylate buffer) for 30 min, and washed again. Dehydration was performed with solutions of increasing ethanol concentration (30%, 50%, 70%, 90%, and 100%) with incubation times of 5–10 min, and the 100% ethanol step was carried out twice. The material was dried using the critical point method in a Leica EM CPD300 equipment (Leica, Wetzlar, Germany). Finally, the samples were coated with gold (Leica EM ACE200) and analyzed in a scanning electron microscope (Jeol JSM6010 Plus/LA, Jeol, Tokyo, Japan) at the Carlos Chagas Institute FIOCRUZ-PR microscopy facility.

### 4.8. Transmission Electron Microscopy of EVs

EVs in PBS were loaded on 300-mesh Formvar-coated copper grids (Electron Microscopy Science, Washington, DC, USA) and allowed to adsorb for 1 h. The grids were washed with PBS and fixed with 2.5% glutaraldehyde (Sigma-Aldrich, St. Louis, MO, USA) in 0.1 M sodium cacodylate buffer (Electron Microscopy Science, Washington, DC, USA) for 10 min. The grids were washed three times with 0.1 M sodium cacodylate buffer and stained with 5% uranyl acetate (Electron Microscopy Science, Washington, DC, USA) for 2 min. After washing with distilled water, the grids were dried and analyzed at an acceleration voltage of 100 kV in a JEOL JEM-1400 Plus transmission electron microscope (JEOL Ltd., Tokyo, Japan) at the Carlos Chagas Institute FIOCRUZ-PR microscopy facility.

### 4.9. Western Blot Analysis of EVs Protein Markers

To increase the protein concentration of EVs, their volume was always concentrated 7.5-fold in Amicon filters (Millipore, Burlington, MA, USA) with a cut-off of 10 kDa. Next, the EVs protein markers profile was characterized by SDS-PAGE and Western blot. Sample buffer containing 4X bromophenol blue, and β-mercaptoethanol (Laemmli buffer) was added to samples (generally 10 to 15 μg of EVs) that were then denatured at 95 °C for 5 min. EV proteins were loaded on a 13% polyacrylamide gel and separated at 20–30 mA in a tank with running buffer (25 mM Tris; 192 mM glycine and 0.1% SDS). The transfer of proteins from the gel to the nitrocellulose membrane (GE Healthcare, Hertfordshire, UK) was carried out at 23 volts for 1 h in a Trans-Blot^®^ SD Semi-Dry Transfer Cell with transfer buffer (25 mM Tris; 192 mM glycine; 20% methanol (*v*/*v*)) chilled in the refrigerator. Proteins in the membrane were identified by staining with 0.2% Ponceau and the membranes were blocked with 5% TBST milk solution (200 mM Tris-HCl (pH 7.5); 1.5 M NaCl; 0.05% of Tween 20; and 5% non-fat dry milk) for 1 h at room temperature. To identify EVs surface markers, the following antibodies were used: anti-CD63 (1:200, Abcam (Cambridge, UK), cat ab59479) and anti-CD9 (1:200, cat. ab236630, Abcam). The antibody anti-TSG101 (1:500, cat. ab125011, Abcam) was used to identify the internal markers of the lumen of extracellular vesicles. Antibodies were diluted in 0.1% TBST-BSA solution. Incubation with primary antibodies occurred overnight under shaking at 4 °C. The membranes were washed thrice under agitation with 0.05% TBST for 5 min. The secondary antibodies used were conjugated with green probe: anti-mouse IgG IRDye Odyssey 800 CW (cat. 926-32210, LI-COR, Lincoln, NE, USA) or anti-rabbit IgG IRDye Odyssey 800 CW (cat. 926-32213, LI-COR). Incubation occurred for 1 h under shaking at room temperature. The membranes were washed under agitation with 0.05% TBST thrice for 5 min. The bands were detected using the image scanner Odyssey Bioanalyzer Infrared Imaging System (LI-COR).

### 4.10. EVs Uptake Assay

For cell/EV interaction evaluation, ADMSC-EVs were labeled with PKH26 Red Fluorescent Cell Linker Kit (cat. PKH26GL, Sigma). The EVs were incubated in the dark under agitation for 10 min with a solution of diluted PKH26 (1:300). After this, the reaction was blocked by adding 3 mL of sterile PBS-BSA 1%. Then, this solution with EVs-PKH26 was subjected to ultracentrifugation for 1 h and 20 min at 100,000× *g* at 4 °C. Then, the supernatant was discarded and the red stained EV pellet was resuspended. These EVs were used in immunofluorescence assays. For this, HUVEC, NHDF-1, and HaCat cells (see Section 4.1) were seeded on coverslips at a concentration of 5 × 10^5^ cells per 1.9 cm^2^, and, after 24 h, cell starvation was performed with a medium change. In the next day, cells were incubated with 24 μg/cover slip of VEs-PKH26 for 48 h. After this period, cells were washed thrice with PBS, fixed with 4% PFA for 20 min at room temperature (RT), and permeabilized and blocked with PBS-BSA1%-TRITON 0.5% for 30 min at RT. Then, the cells were reacted with anti-β-tubulin cytoskeleton marker antibody overnight under shaking. After 16 h, cells were washed again thrice with PBS-BSA1%-TRITON 0.5% and incubated with anti-rabbit Alexa Fluor 488 secondary antibody for 1 h under agitation at RT. Then, cells were stained with DAPI for 20 min at RT for nuclei visualization. These samples were analyzed in Leica TCS SP5 confocal microscope at the Carlos Chagas Institute FIOCRUZ-PR microscopy facility.

### 4.11. Cell Proliferation Assay

Proliferating cells were identified with anti-Ki67 antibody (cat. MA-514520, Invitrogen) in association with DAPI staining. The HUVEC, NHDF-1, and HaCat cells were seeded in an optical 96-well plate at a concentration of 7 × 10^3^ cells per well. After 24 h, starvation was performed with a medium change. The next day, cells were incubated with 8 µg/well with 100 μL medium or 80 ng/μL of VEs-PKH26 for 48 h. After this period, cells were washed thrice with PBS, fixed with 4% PFA for 20 min at room temperature (RT), and permeabilized and blocked with PBS-BSA1%-TRITON 0.5% for 30 min at room temperature. Then, the cells were incubated with anti-Ki67 overnight under shaking followed by anti-rabbit Alexa Fluor 488 secondary antibody staining for 1 h under shaking at RT. Then, cells were washed and incubated with DAPI for 20 min at RT. These samples were analyzed in an Operetta CLS^TM^ High Content Analysis System (Perkin Elmer, Waltham, MA, USA) device. The Harmony^®^ 4.8 High Content Imaging and Analysis Software was used to analyze the data and statistical analysis/graphical representations were generated using the Graphpad Prism 9 software.

### 4.12. Scratch Wound Assay

In order to evaluate cell migration, NHDF-1 and HaCat cells were seeded in a 96-well plate at a concentration of 2 × 10^4^ cells per well. After 24 h, starvation was performed with a medium change. Positive control groups were maintained in FBS supplemented medium. The next day, after starvation, a line was scraped in the middle of each well using a 200 μL tip, and the cells were incubated with 8 μg/well with 100 μL medium or 80 ng/μL of ADMSC-EVs. Then, the plate was immediately incubated in the Operetta High Content device, which was configured to capture photos of the middle well region at 0, 6, 12, 18, and 24 h in order to observe the scratched area and its reduction over time due to cell migration. The data were analyzed and obtained with Harmony^®^ 4.8 High Content Imaging and Analysis Software, and graphics and statistical analysis were generated using the Graphpad Prism 9 software.

### 4.13. Endothelial Tube Formation Assay

HUVEC cells were cultured in EBM-2 medium depleted of FBS one day prior to the seeding in 96-well plates coated with geltrex^TM^ (cat. A1413302, Gibco). HUVEC cells were seeded at a concentration of 5 × 10^4^ cells per well in EBM^®^-2 supplemented with EGM^TM^ 2 MV BulletKit^TM^ (cat. CC-4147, Lonza) and 1% of EV-depleted fetal bovine serum (FBSd). ADMSC-EVs were added at 8 μg/well with 100 μL medium or 80 ng/μL in each experimental replicate. Negative controls were cultured without supplements EGM^TM^ 2 MV and FBSd treated or not treated with ADMSC-EVs. In order to observe any angiogenic potential of ADMSC-EVs, cells were incubated for 19 h and tube formation was observed and photographed with a camera coupled to a Zeiss Microscope (Tokyo, Japan). Images were analyzed with an angiogenesis macro-script in the ImageJ software 1.54i (source code: https://imagej.net/ij/ij/macros/toolsets/Angiogenesis%20Analyzer.txt (accessed on 3 July 2024)).

For immunofluorescence staining, cells were stained with anti-CD31 FITC (cat. 11-0319-42, Invitrogen) or anti-F-actin phalloidin Alexa Fluor 488 (cat. A12379, Invitrogen) and DAPI.

### 4.14. Macrophage Polarization Assay

THP-1 monocytic leukemia cells (cat. 88081201, ECACC, St. Louis, MO, USA) were maintained in endotoxin-free RPMI 1640 Glutamax medium (cat. 72400-047, Gibco) supplemented with 10% FBS at 37 °C in 5% CO_2_ incubator. Cells were subcultured to a concentration of 3 × 10^5^ cells in T-25 culture flask. To differentiate monocytes into macrophages, THP-1 cells were seeded in 24-well plates previously coated with 4 μg/μL of collagen type I solution (cat. C3867-1VL, Sigma) at a concentration of 2 × 10^5^ cells per well in the presence of 50 ng/mL of Phorbol 12-mirystate 13-acetate (PMA; cat. P8139, Sigma) for 48 h. After that, the adherent PMA-primed cells were photographed (bright field), washed twice with RPMI, and incubated in RPMI with 10% FBSd (EV-depleted FBS) for a 24 h resting period. Then, the cells were treated or not treated with ADMSC-EVs and a cocktail of cytokines according to treatment group as follows: (group 1) M0-like macrophages, unstimulated control; (group 2) M0-like macrophages, treated with ADMSC-EVs; (group 3) M1-like macrophages, control; (group 4) M1-like macrophages, treated with ADMSC-EVs; (group 5) M2-like macrophages, control; and (group 6) M2-like macrophages, treated with ADMSC-EVs. Groups treated with ADMSC-EVs (2, 4, and 6) received two subsequential doses of 5 or 15 µg/mL of EVs at days 3 and 4, totaling 10 or 30 µg/mL, respectively (see the experimental design at Appendix A). Groups 3 and 4 of pro-inflammatory M1 macrophages were treated with 20 ng/mL IFN-γ (cat. 300-02, Peprotech, Waltham, MA, USA) and 10 pg/mL LPS on day 3, with incubation lasting until day 6 (72 h of incubation). Groups 5 and 6 of anti-inflammatory M2 macrophages were treated with 20 ng/mL IL-4 (cat. 200-04, Peprotech) and 20 ng/mL IL-13 (cat. 200-13, Peprotech) on day 3, with incubation lasting until day 6 (72 h of incubation). In order to see phenotypic and morphological changes during treatment with PMA, ADMSC-EVs, and cytokines, cells were photographed on days 2 (48 h after seeding and treatment with PMA), 4 (96 h after seeding or 24 h after treatment with cytokines and ADMSC-EVs), and 6 (144 h after seeding or 72 h after treatment with cytokines and ADMSC-EVs) and then immediately submitted to RNA purification (Appendix A).

### 4.15. RNA Isolation, cDNA Synthesis, and Cytokine Expression Evaluation

The RNA was isolated from macrophage lysates from each experimental group using RNeasy mini-kit (QIAGEN, Germantown, MD, USA) with DNAse treatment according to the manufacturer’s instructions. cDNAs were synthesized from isolated RNAs through reverse transcription using ImProm-II™ Reverse Transcription System kit (cat. A3800, Promega, Tokyo, Japan) according to the manufacturer’s manual. Briefly, for each sample, 1 µg of RNA was mixed with 0.5 µg/µL oligodT and nuclease-free water, and then incubated at 70 °C for 5 min. Then, samples were incubated in iced bath for 5 min. A mix solution containing Buffer 5X (cat. M289C, Promega), 25 mM MgCl_2_ (cat. A351H, Promega), 10 mM dNTPs, 40 U/µL RNAseout (cat. 100000840, Thermo, Waltham, MA, USA), and 160 U/µL Improm II reverse transcriptase (cat. M314C, Promega) was added to each sample. Finally, samples were incubated at 25 °C for 5 min, 42 °C for 60 min, and 70 °C for 15 min. Samples were incubated in ice, diluted in nuclease-free water, and stored at −20 °C.

### 4.16. Relative Expression of Cytokines

cDNA samples of M0, M1-like, and M2-like macrophages were submitted to qPCR using primers for the following genes: IL-1β (primers sense 5′-ATGGCTTATTACAGTGGCA-3′ and antisense 5′-GTAGTGGTGGTCGGAGATT-3′) [112]; TNF-α (primers sense 5′-CCCAGGGACCTCTCTCTAATC-3′ and antisense 5′-ATGGGCTACAGGCTTGTCACT-3′) [112]; IL-10 (primers sense 5′-TCAAGGCGCATGTGAACTCC-3′ and antisense 5′-GATGTCAAACTCACTCATGGCT-3′) [112]; TGF-β (primers sense 5′-GAAACCCACAACGAAATCTATGAC-3′ and antisense 5′-ACGTGCTGCTCCACTTTTAACT-3′) [113]; and HPRT (primers sense 5′-GACCAGTCAACAGGGGACAT-3′ and antisense 5′-CCTGACCAAGGAAAGCAAAG-3′). Each reaction consisted of 5 µL of GoTaq^®^ qPCR Master Mix 2× (cat. A600A, Promega), 0.5 µL of Forward and Reverse primers (10 pmol/µL), 1 µL of RNAse-free water, and 3 µL of each cDNA, making up a total of 10 µL. Reactions were performed at 95 °C for 2 min, then cycling was carried out and repeated 40 times at 95 °C for 15 s and 60 °C for 60 s. Then, a melting curve was carried out at 95 °C for 15 s, 60 °C for 60 s, and 97 °C for 1 s in the last incubation. Results were analyzed with Quant studio 5 Software v.3.5 and the relative gene expression was calculated using the 2^−ΔΔCT^ method, as previously described [114]. All the groups were compared to unstimulated M0 macrophages (group 1).

### 4.17. Statistical Analysis

Statistical analyses were performed using GraphPad Prism 9.0 software. Appropriate tests were used to calculate statistical significance in each experiment. For the proliferation assay and macrophage cytokine expression analysis, involving the comparison of two groups (EV-treated and non-EV-treated), a two-way ANOVA with Šídák’s correction was performed, with a confidence interval of 95%. For the angiogenesis assay, an unpaired nonparametric Kolmogorov–Smirnov was used for comparisons between cells treated or not treated with EVs, with a confidence interval of 95%. For the migration experiments, involving comparison of two groups, a paired Student’s *t*-test was first performed with a confidence interval of 95%. Differences among groups were evaluated by two-way ANOVA with Geisser–Greenhouse correction, and 95% confidence intervals were calculated. All data are reported as the mean ± standard deviation. Differences were considered statistically significant at * *p* < 0.05 and ** *p* < 0.01.

## 5. Conclusions

In this work, it was shown that ADMSC-EVs induce the proliferation of HaCat, the migration of HaCat and NHDF-1, tube-like structure formation in HUVEC cultures, and an increase in the expression of IL-1β and TGF-β genes in M1 macrophages, roles that could help in wound repair. In addition, the use of cytodex-1 is suitable for the expansion of ADMSCs and the isolation and application of EVs in cell culture. Therefore, ADMSC-EVs isolated from a scalable cell suspension culture are suitable for in vitro application and able to induce different cell responses. The regenerative and anti-inflammatory potentials of ADMSC-EVs from suspension cultures remains to be further investigated in vivo.

## Figures and Tables

**Figure 1 ijms-25-07605-f001:**
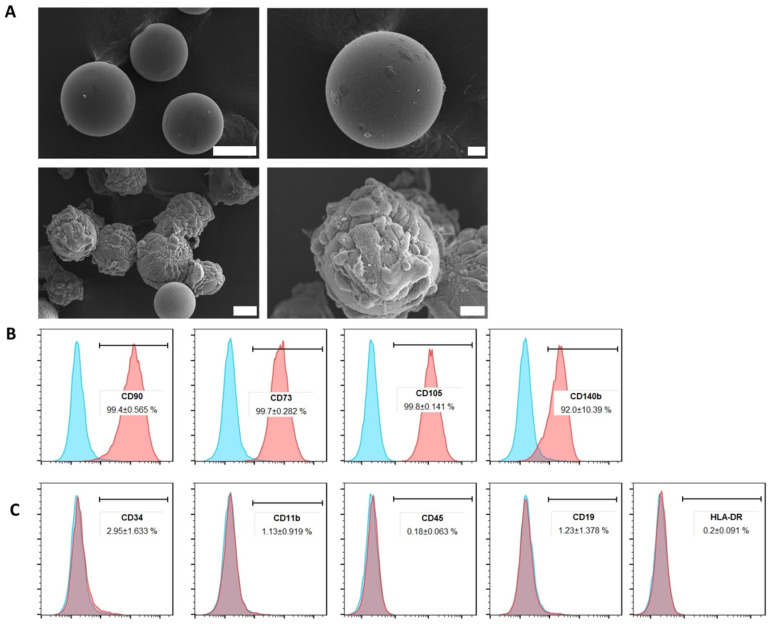
Growth characterization and immunophenotyping of ADMSCs cultured on cytodex-1. ADMSCs were cultured on cytodex-1 for 4 days until immunophenotyping. (**A**) Scanning electron microscopy (SEM) of cytodex-1 microcarriers. Upper panel shows empty microcarriers without attached cells. Lower panel shows ADMSCs in passage 8 plated in a concentration of 5 × 10^5^ cells to 1 g/L of cytodex-1 microcarriers. Scale bar in upper images from left to right = 50 and 10 µm, and in lower images from left to right = 50 and 20 μm. (**B**,**C**) Representative histograms of cytodex-1-cultured ADMSC immunophenotypic profile: (**B**) positive and (**C**) negative markers. In all graphs, the blue and the red peaks represent the isotypic control and the tested cell sample respectively. Representative figures of two independent assays. Data are reported as the mean ± standard deviation.

**Figure 2 ijms-25-07605-f002:**
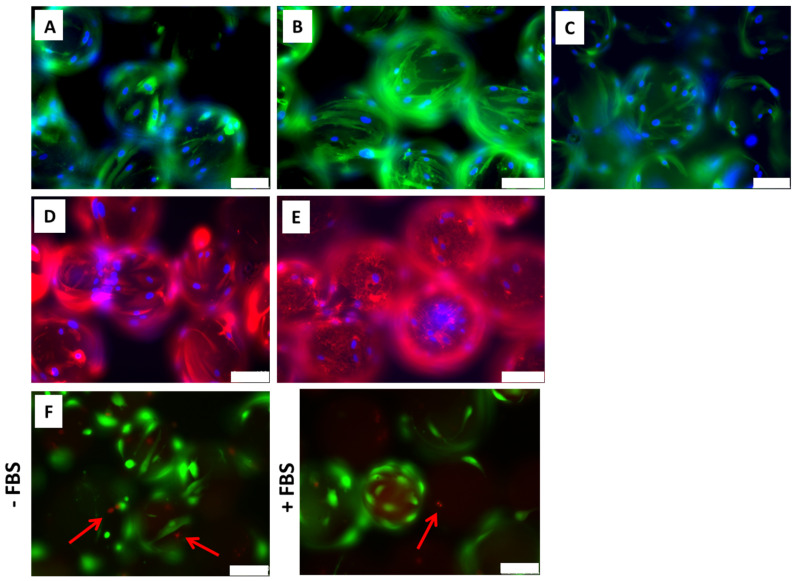
Characterization of cytoskeleton, extracellular matrix, and cell viability of ADMSCs cultivated on cytodex-1 microcarriers. Immunolabeling of cytoskeletal proteins: (**A**) anti-β-actin (green); (**B**) phalloidin F-actin probe (green); (**C**) anti-β-tubulin (green); and (**D**) anti-vimentin (red). (**E**) Labeling of the extracellular matrix protein fibronectin (red) secreted by ADMSCs cultured on the cytodex-1 microcarriers. (**A**–**E**) Nuclei were stained with DAPI (blue). (**F**) Cellular viability of ADMSCs cultured on cytodex-1 in medium depleted of FBS or not depleted of FBS. Representative images show live cells stained in green and dead cells with nuclei stained in red (red arrows). Scale bars = 100 µm.

**Figure 3 ijms-25-07605-f003:**
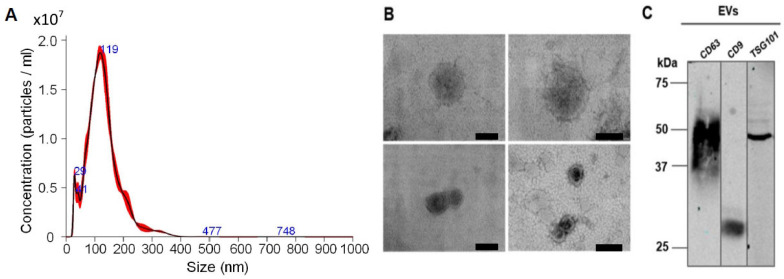
ADMSC-EVs characterization. (**A**) Representative graph showing size and concentration of EVs analyzed by NTA. (**B**) Transmission electron microscopy (TEM) representative images of ADMSC-EVs showing membrane bilipid layer. Scale bar = 50 nm, except for the lower image at right with scale bar = 100 nm. (**C**) Characterization of protein markers of ADMSC-EVs by Western blot showing the presence of CD63 (~55 kDa), CD9 (25 kDa), and TSG101 (46 kDa) proteins. Representative image of three independent isolations.

**Figure 4 ijms-25-07605-f004:**
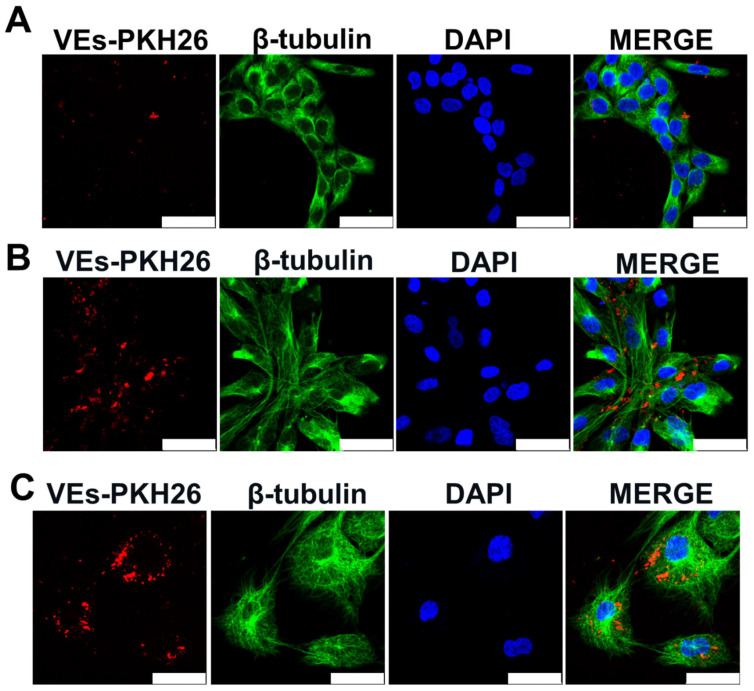
ADMSC-EVs are taken up by HaCat (**A**), NHDF-1 (**B**), and HUVEC (**C**). Representative confocal microscopy images showing that cells incubated with 24 μg/mL EVs-PKH26 (vesicles stained with the red stain PKH26) for 48 h are capable of being taken up. Cytoskeletons of cells are labeled with anti-β-tubulin FITC (green). Nuclei are stained with DAPI (blue). Scale bar = 50 μm.

**Figure 5 ijms-25-07605-f005:**
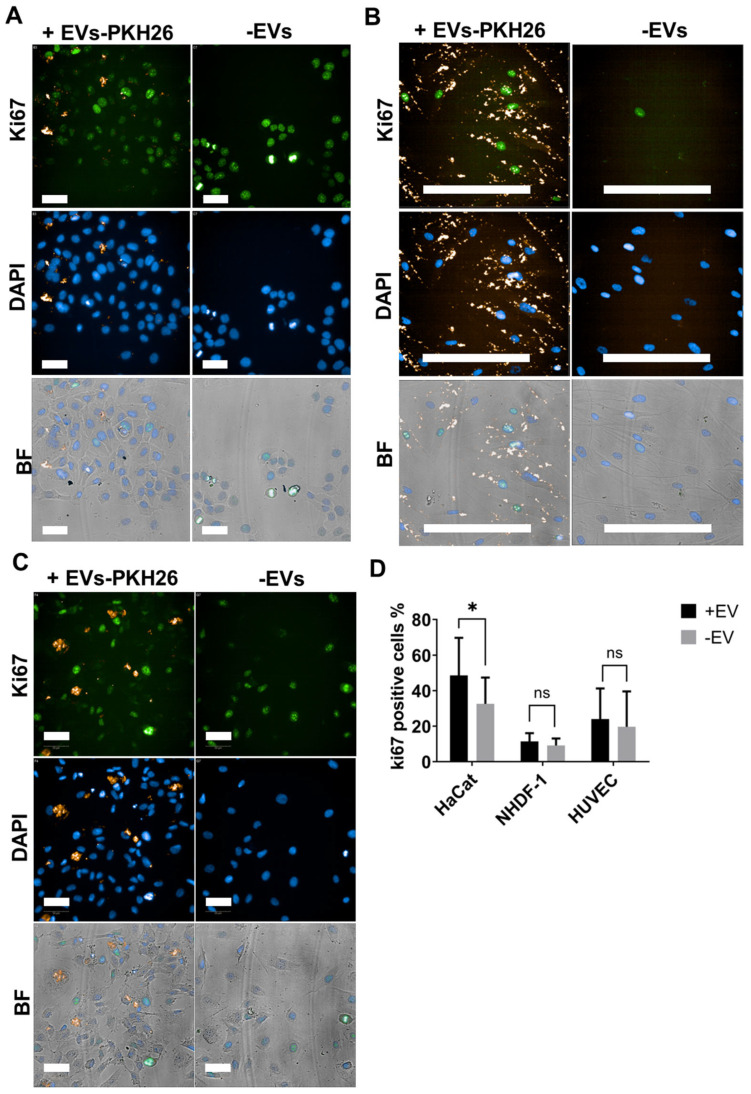
ADMSC-EVs induced keratinocyte proliferation. Representative images of Ki67 assay of HaCat (**A**), NHDF-1 (**B**), and HUVEC (**C**) cells treated (+) or not treated (−) with 80 ng/µL ADMSC-EVs. Immunofluorescence was performed with anti-Ki67 and a secondary antibody conjugated with Alexa Fluor 488 (green). Nuclei were stained with DAPI (blue). EVs were stained with PKH26 (seen in yellow). Largest scale bar = 200 µm, and smallest scale bar = 50 µm. (**D**) Proliferation graph of four independent assays (with six replicates each), showing groups treated or not treated with ADMSC-EVs. For a comparison of the two groups, a two-way ANOVA with Šidák correction and a confidence interval of 95% was performed, and treatments were statistically significant at *p* < 0.05 *. ns = not statistically significant.

**Figure 6 ijms-25-07605-f006:**
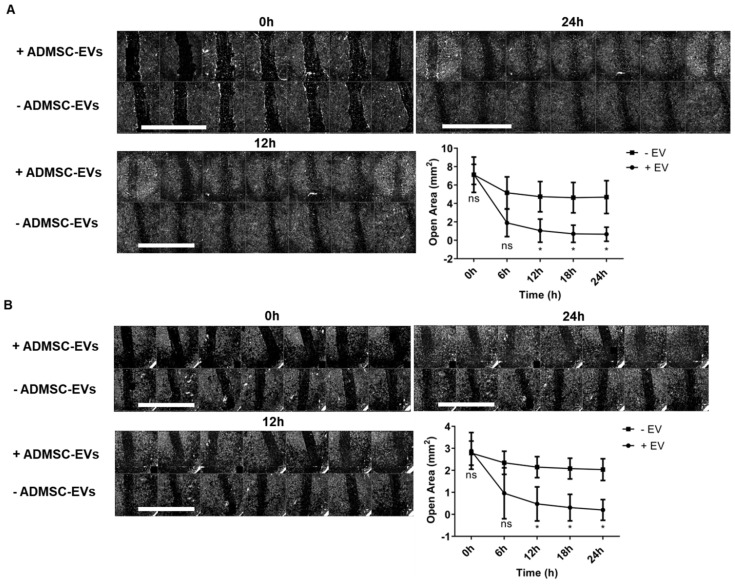
ADMSC-EVs induced fibroblast and keratinocyte migration. (**A**) Representative images of seven wells (lines) of NHDF-1 migration at 0, 12, and 24 h and graph of scratch/wound closure at 0, 6, 12, 18, and 24 h treated (+) or not treated (−) with 80 ng/µL ADMSC-EVs. (**B**) Representative images of seven wells (lines) of HaCat migration at 0, 12, and 24 h and graph of scratch/wound closure at 0, 6, 12, 18, and 24 h treated (+) or not treated (−) with 80 ng/µL ADMSC-EVs. Data were obtained from three independent assays, with seven wells for each condition, and without FBS supplementation. For the comparison of two groups, a paired Student’s *t*-test with a confidence interval of 95% was performed. Differences among groups were evaluated by two-way ANOVA with Geisser–Greenhouse correction, and 95% confidence intervals were calculated. Differences were considered statistically significant at *p* < 0.05 *. ns = not statistically significant. Scale bar = 5 mm.

**Figure 7 ijms-25-07605-f007:**
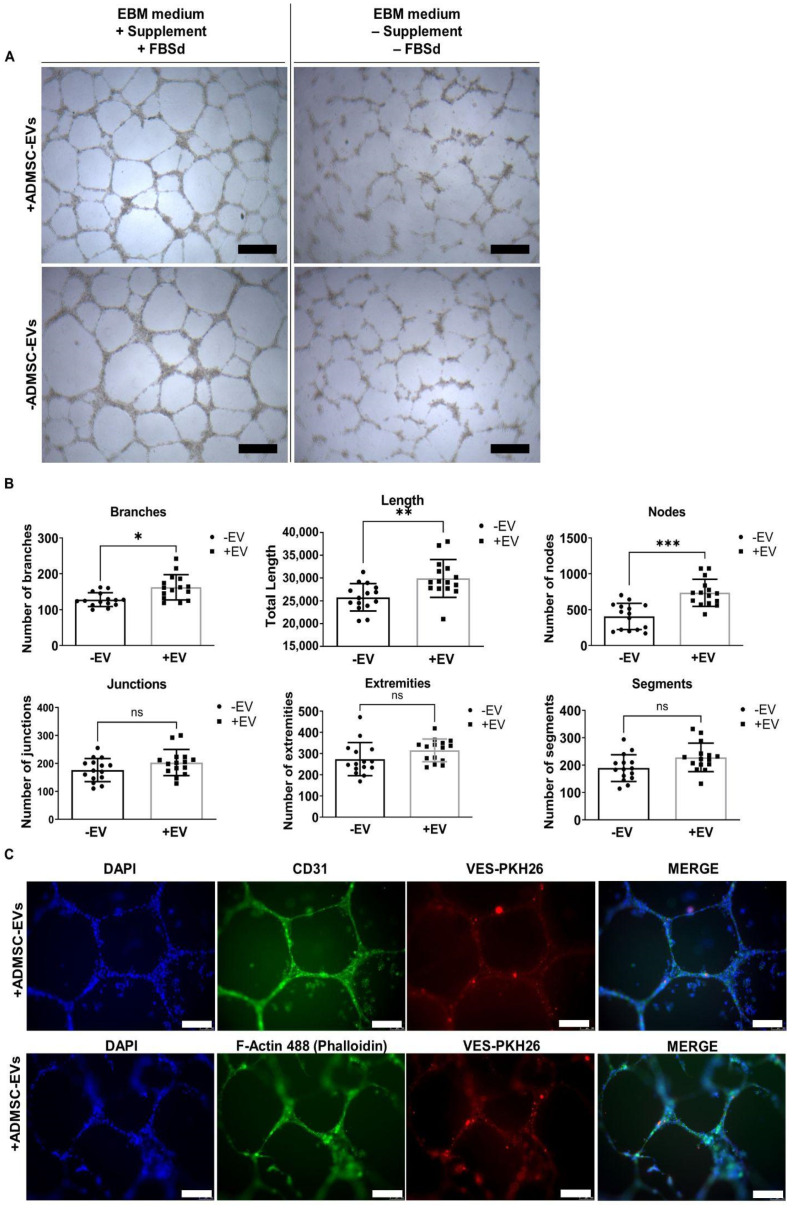
ADMSC-EVs induced angiogenesis. (**A**) Organization of tube-like structures in HUVECs treated (+) or not treated (−) with 80 ng/µL ADMSC-EVs for 24 h in EBM medium supplemented with FBSd (EV-depleted FBS) and EGM-2 supplements or in raw EBM medium depleted of supplements and FBS. (**B**) Quantitative graphs of number of extremities, branches, junctions, nodes, segments, and length of branches in HUVEC cultures with/without ADMSC-EVs treatment. Graphs of five independent assays (with triplicates each). An unpaired nonparametric Kolmogorov-Smirnov test was used for comparisons between cells with/without ADMSC-EVs treatment, within a confidence interval of 95%. Differences were considered statistically significant at * *p* < 0.05, ** *p* < 0.01, and *** *p* < 0.001. (**C**) Immunofluorescence imagens of tube-like structures in cells treated (+) with 80 ng/µL with EVs-PKH26 (red) for 24 h and maintained in EBM medium with supplements and FBSd. Cells were stained for endothelial marker CD31 (green) or F-actin phalloidin 488 (green). Nuclei were stained with DAPI (blue). ns = not statistically significant. Scale bar = 100 µm.

**Figure 8 ijms-25-07605-f008:**
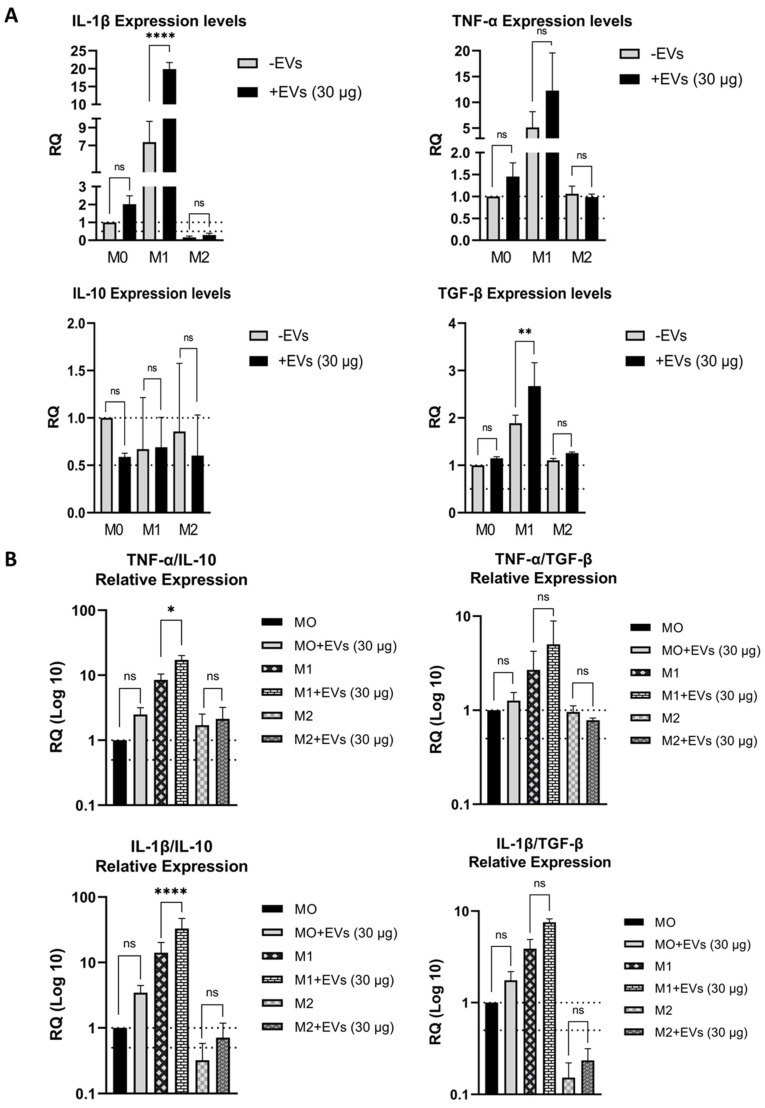
Higher concentration of ADMSC-EVs induced IL-1β and TGF-β expression in M1-like macrophages. Graphs of cytokine expression on M0, M1-like, and M2-like macrophages treated (+) or not (−) with 30 μg/mL (**A**) ADMSC-EVs. Graphs of ratios between pro-inflammatory and anti-inflammatory cytokine expression in M0, M1, and M2 macrophages treated (+) or not (−) with 30 μg/mL (**B**) ADMSC-EVs. Data obtained from three independent assays with triplicates each. Relative gene expression quantification (RQ) was calculated using the 2^−ΔΔCT^ method. The dotted lines indicate the 2^−ΔΔCT^ value between 1 and 0.5 to facilitate visualization in relation to gene expression without change (equal to 1) and reduced by half (equal to 0.5). A two-way ANOVA with Šídák’s correction was used for comparisons between cells treated and non-treated with EVs, within a confidence interval of 95%. Differences were considered statistically significant at * *p* < 0.05, ** *p* < 0.01 and **** *p* < 0.0001. ns = not statistically significant.

## Data Availability

The raw data supporting the conclusions of this article will be made available by the authors on request.

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
