# Peer review of "Cellular In Vitro Responses Induced by Human Mesenchymal Stem/Stromal Cell-Derived Extracellular Vesicles Obtained from Suspension Culture"

_ijms, 2024, doi:10.3390/ijms25147605_

Round 1

Reviewer 1 Report

Comments and Suggestions for Authors

In this manuscript, the authors present substantial evidence demonstrating that EVs from microcarrier-cultivated ADMSCs can modulate cellular behaviors, such as proliferation and migration, by balancing pro-inflammatory and anti-inflammatory responses in macrophages. Overall, these findings are interesting and suggest that EVs may also induce pro-inflammatory responses under certain conditions. However, some sections of the manuscript are overly lengthy, and the discussion could be further expanded. Thus, a major revision is recommended.

Specific comments are as follows:

1. The abstract contains excessive background information. It is advisable to transfer some content to the introduction to sharpen the focus on the manuscript's main findings and its unique contributions.

2. Line 61, it is unclear whether native MSC EVs carry pro-inflammatory or anti-inflammatory cytokines. Could the authors clarify this?

3. Dose lacking FBS supplementation increase apoptotic bodies in EVs, consequently increasing inflammatory factors within EVs?

4. The manuscript notes that parental cells were cultured under 3D conditions and subjected to starvation during EV collection. The reviewer recommends that the authors include a more detailed discussion on how external stimuli on donor cells might explain the pro-inflammatory responses induced by EVs. For instance, previous research has identified other external stimuli such as hypoxia (doi.org/10.1038/s41598-022-05161-7) and electroporation (doi.org/10.1002/advs.202302622) as factors. Please include these references or more recent studies to enrich the discussion on this topic.

Author Response

Dear reviewer,
We are submitting the revised version of our manuscript for your consideration. Thank you for your careful review of our manuscript. We also appreciate all your comments and suggestions, which have been addressed point-by-point in the attached document.

Kind regards!

Reviewer 2 Report

Comments and Suggestions for Authors

This manuscript deals with the ability of extracellular vesicles isolated from adipose mesenchymal/stromal cells (ADMSC) to affect some functional activities of keratinocytes, endothelial cells, THP1 tumor cell line, and fibroblasts.

Overall the experiments are well performed.

The message that ADMSC can affect some functions of the cells used is either expected or already well-known. The main point of originality for this reviewer is the mode of culture of the ADMSC.

However, a comparison between vesicles produced in conventional conditions and on sepharose beads (as a 3D model) has not been performed.

The content of the microvesicles is not tested. It is well known that EGF, VEGF. IL6 and other cytokines and factors can be present in extracellular vesicles from ADMSC or MSC in general. Also, microvesicles can contain other components such miRNA and others that may regulate several functions. This should be considered to better characterize the microvesicles.

The test to evaluate the proliferation should be not only the Ki67 expression but also the ATP content and the CV or MTT assay.

The differentiation of THP1 cells to Mo, M1, and M2 should be shown in the main manuscript to confirm the supposed effect of PMA. 

There are several limitations of this manuscript such as:

1-microvesicles are derived from 1 donor of adipose mesenchymal stromal cells (ADMSC). The findings reported are true for just this ADMSC donor of microvesicles.

2- The target cells of the microvesicles are again just one example of target cells.

3- THP1 cells are not healthy monocytes. The findings reported are true for this specific cell line.

4- The finding that microvesicles from ADMSC can affect some properties of the target cells listed is highly expected of even already known. That EGF and VEGF as well as other factors can be produced by MSC is well-known.

I think that the authors should test several ADMSC from different donors, and the same should be done with target cells. THP1 cells should be substituted with peripheral blood monocytes from healthy donors.

Without these data, the experiments are strictly true for the cell population considered in this work.

I know very well that several authors consider THP1 and monocytes but this is not correct at all. I would say they are a model, but the use of microvesicles is supposed not in a model but in humans.

On the other hand, the mode of culture is interesting if applied to several donors of ADMSC while the information given in general is actually expected.

Minor points

The introduction is too long and also the abstract is about half on what is already well-established.

Comments on the Quality of English Language

English is good.

Author Response

(The authors gave the same response as above.)

Round 2

Reviewer 1 Report

Comments and Suggestions for Authors

The authors have satisfactorily addressed my concerns. Well done!

Author Response

We thank the reviewer for the time and attention with our demand.

Reviewer 2 Report

Comments and Suggestions for Authors

The authors revised the manuscript but they did not perform the experiments suggested.

I understand the explanations but actually the message of this manuscript should be improved analysing more than one ADMSC  or with just THP1 as a model for macrophages. In this form I cannot endorse the manuscript for publication.

I do not have any conflict of interest with this manuscript, the authors or its content.

Comments on the Quality of English Language

English is good

Author Response

Dear reviewer, we are very sorry about your decision. We believe we have unequivocally demonstrated (using experimental triplicates and a number of 5 to 7 replicates within each experiment) that the cultivation model proposed in our manuscript is viable and capable of providing quality EVs. We are certain that the message of the manuscript is important for the scientific community studying the use of EVs in regenerative medicine protocols. We agree on the importance of analyzing more than one ADMSC and other experimental models, this will be our next step. However, we believe that the path would not be through repeating these experiments to confirm the current data. The current results already point to the viability of the cultivation model for the isolation of EVs in large quantities, which will enable the application of these active EVs in other more representative experimental models, such as in vivo assays, whose viability would be compromised without the standardization of the cell culture method used in our manuscript. We respectfully thank you for your time and attention.